# Food additive mixtures and type 2 diabetes incidence: Results from the NutriNet-Santé prospective cohort

**Marie Payen de la Garanderie**[1,2*], **Anaïs Hasenbohler**[1,2], **Nicolas Dechamp**[1], **Guillaume Javaux**[1,2], **Fabien Szabo de Edelenyi**[1], **Cédric Agaësse**[1], **Alexandre De Sa**[1], **Laurent Bourhis**[1], **Raphaël Porcher**[3], **Fabrice Pierre**[2,4], **Xavier Coumoul**[2,5], **Emmanuelle Kesse-Guyot**[1,2], **Benjamin Allès**[1], **Léopold K. Fezeu**[1], **Emmanuel Cosson**[1,6], **Sopio Tatulashvili**[1,6], **Inge Huybrechts**[7], **Serge Hercberg**[1,2,8], **Mélanie Deschasaux-Tanguy**[1,2], **Benoit Chassaing**[2,9], **Héloïse Rytter**[9], **Bernard Srour**[1,2], **Mathilde Touvier**[1,2*]

1 Université Sorbonne Paris Nord and Université Paris Cité, INSERM, INRAE, CNAM, Center of Research in Epidemiology and StatisticS (CRESS), Nutritional Epidemiology Research Team (EREN), Bobigny, France, 2 Network for Nutrition And Cancer Research, Jouy-en-Josas, France, 3 Centre d'Épidémiologie Clinique, AP-HP, Hôtel-Dieu, Paris, France, 4 Toxalim (Research Centre in Food Toxicology), Université de Toulouse, INRAE, ENVT, INP-Purpan, UPS, Toulouse, France, 5 INSERM UMR-S 1124, Université de Paris, Paris, France, 6 Diabetology, endocrinology and nutrition Department, Avicenne Hospital, AP-HP, Bobigny, France, 7 International Agency for Research on Cancer, World Health Organization, Lyon, France, 8 Public Health Department, Groupe Hospitalier Paris-Seine-Saint-Denis, Assistance Publique-hôpitaux de Paris (AP-HP), Bobigny, France, 9 Microbiome–Host Interactions, Institut Pasteur, Université Paris Cité, INSERM U1306, Paris, France

* m.payen@eren.smbh.univ-paris13.fr (MPDLG); m.touvier@eren.smbh.univ-paris13.fr (MT)

## Abstract

### Background

Mixtures of food additives are daily consumed worldwide by billions of people. So far, safety assessments have been performed substance by substance due to lack of data on the effect of multiexposure to combinations of additives. Our objective was to identify most common food additive mixtures, and investigate their associations with type 2 diabetes incidence in a large prospective cohort.

### Methods and Findings

Participants ($n$ = 108,643, mean follow-up = 7.7 years (standard deviation (SD) = 4.6), age = 42.5 years (SD = 14.6), 79.2% women) were adults from the French NutriNet-Santé cohort (2009–2023). Dietary intakes were assessed using repeated 24h-dietary records, including industrial food brands. Exposure to food additives was evaluated through multiple food composition databases and laboratory assays. Mixtures were identified through nonnegative matrix factorization (NMF), and associations with type 2 diabetes incidence were assessed using Cox models adjusted for potential socio-demographic, anthropometric, lifestyle and dietary confounders. A total of 1,131 participants were diagnosed with type 2 diabetes. Two out of the five identified food additive mixtures were associated with higher type 2 diabetes incidence: the first mixture included modified starches, pectin,

**Data availability statement:** Researchers from public institutions can submit a request to have access to the data for strict reproducibility analysis (systematically accepted) or for a new collaboration, including information on the institution and a brief description of the project to collaboration@etude-nutrinet-sante.fr. All requests will be reviewed by the steering committee of the NutriNet-Santé study. If the collaboration is accepted, a data access agreement will be necessary and appropriate authorizations from the competent administrative authorities may be needed. In accordance with existing regulations, no personal data will be accessible. R/SAS code is available without restrictions upon request at collaboration@etude-nutrinet-sante.fr.

**Funding:** The NutriNet-Santé study was supported by the following public institutions : Ministère de la Santé, Santé Publique France, Institut National de la Santé et de la Recherche Médicale (INSERM), Institut National de la Recherche pour l'agriculture, l'alimentation et l'environnement (INRAE), Conservatoire National des Arts et Métiers (CNAM), and University Sorbonne Paris Nord. This project has received funding from the European Research Council (ERC) under the European Union's Horizon 2020 research and innovation program (grant agreement No 864219, ADDITIVES), the French National Cancer Institute (INCa_14059), the French Ministry of Health (arrêté 29.11.19) and the IdEx Université de Paris (ANR-18-IDEX-0001), and a Bettencourt-Schueller Foundation Research Prize 2021. This project was awarded the NACRe (Network for Nutrition and Cancer Research) Partnership Label. BC's laboratory is supported by a Starting Grant from the European Research Council (ERC) under the European Union's Horizon 2020 research and innovation program (grant agreement No. ERC-2018-StG- 804135 INVADERS), and the national program "Microbiote" from INSERM. This work only reflects the authors' view, and the funders are not responsible for any use that may be made of the information it contains. Researchers were independent from funders. The funders had no role in study design, data collection and analysis, decision to publish, or preparation of the manuscript.

**Competing interests:** The authors have declared that no competing interests exist.

guar gum, carrageenan, polyphosphates, potassium sorbates, curcumin, and xanthan gum (hazard ratio (HR)$_{\text{per an increment of 1SD of the NMF mixture score}}$ = 1.08 [1.02, 1.15], $p = 0.006$), and the other mixture included citric acid, sodium citrates, phosphoric acid, sulphite ammonia caramel, acesulfame-K, aspartame, sucralose, arabic gum, malic acid, carnauba wax, paprika extract, anthocyanins, guar gum, and pectin (HR = 1.13 [1.08,1.18], $p < 0.001$). No association was detected for the three remaining mixtures: HR = 0.98 [0.91, 1.06], $p = 0.67$; HR = 1.02 [0.94, 1.10], $p = 0.68$; and HR = 0.99 [0.92, 1.07], $p = 0.78$. Several synergistic and antagonist interactions between food additives were detected in exploratory analyses. Residual confounding as well as exposure or outcome misclassifications cannot be entirely ruled out and causality cannot be established based on this single observational study.

## Conclusions

This study revealed positive associations between exposure to two widely consumed food additive mixtures and higher type 2 diabetes incidence. Further experimental research is needed to depict underlying mechanisms, including potential synergistic/antagonist effects. These findings suggest that a combination of food additives may be of interest to consider in safety assessments, and they support public health recommendations to limit nonessential additives.

## Trial Registration

The NutriNet-Santé cohort is registered at clinicaltrials.gov (NCT03335644). https://clinicaltrials.gov/study/NCT03335644.

---

## Author summary

### Why was this study done?

- Several experimental and epidemiological studies have suggested potential deleterious effects of some food additives widely used by the food industry to enhance the texture, shelf life, taste, and appearance of foods.

- So far, research and safety evaluation of food additives has been conducted on a substance-by-substance basis, while in real-life, food additive mixtures are consumed by billions of people globally.

- Some experimental studies have raised concerns about potential interactions between additives within mixtures and their potential impact on health, but human epidemiological data are lacking.

### What did the researchers do and find?

- In this large cohort of 108,643 French adults, five frequently consumed food additive mixtures were identified.

- Two of them were associated with higher type 2 diabetes incidence, independently of the nutritional quality of the diet, and after adjustment for a wide range of potential confounders: the first mixture primarily consisted of emulsifiers, preservatives, and a dye,

**Abbreviations:** EFSA, European Food Safety Authority; GNPD, Global New Products Database; HR, hazard ratio; IPAQ, International Physical Activity Questionnaire; GSFA, General Standard for Food Additives; NMF, nonnegative matrix factorization; OQALI, Observatoire de la Qualité de l'Alimentation; SD, standard deviation; UPF, ultra-processed foods; 95% CIs, 95% confidence intervals.

while the second mixture was characterized by acidifiers, acid regulators, dyes, artificial sweeteners, and emulsifiers.

- Exploratory analyses suggested both synergistic and antagonist interactions between several food additives emblematic of these mixtures.

## What do these findings mean?

- To our knowledge, this study is the first to estimate the exposure to food additive mixtures in a large population-based cohort and investigate their link with type 2 diabetes incidence. These results suggest that food additives found in a wide variety of products and frequently consumed together may potentially represent a modifiable risk factor for type 2 diabetes prevention. They support public health recommendations to limit nonessential additives.

- The potential synergisms and antagonisms may be of interest in future mechanistic investigations, to better understand the relative influence of individual additives and their interactions in the observed associations.

- Main limitations include possible exposure and outcome measurement errors and the fact that causality cannot be established on the basis of this observational study alone.

## Introduction

Ultra-processed foods (UPF) are endemic in Western diets and represent between 15%–20% (e.g., in Columbia, Romania) to almost 60% (in the United States) of daily energy intake [1]. Mounting evidence from epidemiological and experimental studies suggests a deleterious impact of UPF on many health outcomes, in particular metabolic-related diseases [1,2]. Beyond their poorer nutritional quality on average, one of the hypotheses to explain the health effects of UPF is the large use of food additives by the industry [1].

The World Health Organization defines food additives as substances primarily added to foods on an industrial scale, for technical purposes (e.g., lengthened product's shelf-life, improve texture, taste, color, and palatability) [3]. In Europe, > 300 food additives are authorized (e.g., emulsifiers, artificial sweeteners, colors, preservatives) and their use in food manufacturing is governed by European regulation EC/1333/2008. Their safety has previously been assessed by the European Food Safety Authority (EFSA), which proposed acceptable daily intake for some of them.

However, these evaluations were constrained by the available scientific evidence at the time, which was limited due to a lack of human data and a predominant focus on specific toxicological targets such as cytotoxicity and genotoxicity. Recent in vitro/in vivo experimental studies now suggest deleterious effects of some food additives on a wider spectrum of health outcomes, including metabolic disorders, chronic inflammation, and gut microbiota disruption leading to intestinal inflammation [4]. Moreover, the NutriNet-Santé cohort study, which collected unique detailed dietary exposure data, including commercial names and brands of industrial products, provided new human data insights, suggesting associations between dietary exposure to widely consumed food additives (e.g., some artificial sweeteners and emulsifiers) and higher incidence of several chronic diseases, in particular type 2 diabetes [5,6].

Another important gap to date has been that previous evaluations have not been able to account for potential interaction/synergistic effects when assessing the safety of additives due

to a lack of data. Single UPF often contain mixtures of additives [7]. Moreover, diets rich in UPF lead to the consumption of food combinations that result in the ingestion of mixtures of food additives. These additives may interact through synergistic or antagonistic effects, potentially influencing metabolism and overall health [8,9]. In a recent in vitro study based on four human cell models, we observed toxicological effects food additive mixtures, beyond the effect of these substances alone [10].

To our knowledge, this study is the first to aim at identifying the main mixtures of food additives and studying their associations with type 2 diabetes incidence using the large prospective NutriNet-Santé cohort. An exploratory, secondary aim was to examine interactions between food additives in mixtures associated with type 2 diabetes incidence, to explore potential synergisms and/or antagonisms.

## Methods

### Study population

This study was conducted within the population-based NutriNet-Santé prospective e-cohort. This French study was launched on May 11th 2009 with an ongoing open enrollment of volunteers. Its main objective is to investigate the relationships between nutrition and health [11]. Participants are recruited through vast multimedia campaigns from the general population of French citizens aged ≥15 years with internet access. To enroll, they are required to create a personal account on the NutriNet-Santé web-based platform (https://etude-nutri-net-sante.fr/). All participants enrolled up to December 31st, 2023 were included in this study. Upon enrollment, participants are invited to provide detailed information by completing five questionnaires about their lifestyle and socio-demographic data (e.g., date of birth, sex, educational level, professional occupation, smoking status, number of children), health status (e.g., personal and family medical history, medical treatments), dietary habits (three nonconsecutive 24-h dietary records [12]), anthropometric data (e.g., height, weight), and physical activity level (7-day assessment via the International Physical Activity Questionnaire [IPAQ]) [13]. Ethnicity and religion were not recorded since these variables are considered sensitive data by the French law, which strictly regulate their collection in population-based studies.

### Ethical approval

NutriNet-Santé is registered at ClinicalTrials.gov (NCT03335644), conducted according to the Declaration of Helsinki guidelines, and approved by the Institutional Review Board of the French Institute for Health and Medical Research (IRB-Inserm) and the "Commission Nationale de l'Informatique et des Libertés" (CNIL n°908450/n°909216). Each participant provides an electronic informed consent prior to enrollment.

### Dietary data collection

At inclusion, and every 6 months thereafter, participants were invited to fill out three non-consecutive days of 24-h dietary records, randomly assigned over a 2-week period, including 2 weekdays and 1 weekend day (to account for variability in the diet across the week and the seasons[14–16]. Details on the dietary data collection and energy under-reporter's (i.e., participants who systematically reported implausibly low energy intakes) identification are provided in eMethod1 in S1 Appendix. We calculated daily dietary intakes for food additives, nutrients, energy and food groups as the mean intake from all 24-h dietary records available for each participant during their first two years of follow-up. The NOVA classification was applied to identify UPF and calculate their contribution to energy intake [17].

## Food additive intakes

Intakes of food additives were quantified based on data provided by the participants' dietary records, in which the commercial brand/name of the industrial products consumed were recorded. The presence/absence of each specific additive in each specific food was determined by a dynamic matching with several databases, considering the date of consumption (to account for reformulations across time). Multiple sources were used to retrieve food additive doses (including ad hoc laboratory assays and EFSA data). The detailed method for estimating food additive intakes was previously described [8]; more information is provided in eMethod2 in S1 Appendix. Table A in S1 Appendix displays the list of the 269 food additives ingested by the participants with corresponding EU codes. In order to obtain a reliable estimate of food additive exposure and to focus on those most likely to have substantial public health impact, only those consumed by at least 5% of the cohort were included in the mixture modeling.

## Type 2 diabetes ascertainment

Type 2 diabetes was assessed using a multisource approach. Throughout the follow-up period, participants were invited to report any health events, medical treatments, and examinations via the biannual health questionnaires or at any time, directly via the health interface of their personal account. Furthermore, the NutriNet-Santé cohort was linked to the national health insurance system database in order to obtain additional information regarding medical treatments and consultations. Linkage to the French National Mortality Registry (CépiDC) enabled the identification of the occurrence and cause of death. Further details can be found in eMethod3 in S1 Appendix.

## Statistical analyses

Among participants from the NutriNet-Santé cohort who completed at least two 24-h dietary records during their first 2 years of follow-up, we included those who were not under-energy reporters and who did not have any prevalent type 1 or 2 diabetes diagnosed before their enrollment in the cohort. Participants with a null food additive consumption ($n = 80$, 0.07%) were excluded to perform mixture analyses (flowchart of participants presented in Fig A in S1 Appendix). Food additive mixtures were identified using nonnegative matrix factorization (NMF). This size reduction technique was specifically adapted to sparse matrices containing positive values [18]. The Lee algorithm was selected for its ability to balance residual minimization and sparseness [19], while ensuring nonnegativity and effectively handling noise. The number of ranks (i.e., the number of NMF components to retain) was determined according to the method proposed by Brunet and colleagues [20], using the smallest number for which the cophenetic coefficient starts decreasing, as visualized on the consensus map (Table C.a. in S1 Appendix). The NMF was performed using the R package NMF [21] and the scores arising from the components (= the food additive mixtures) were scaled. Food additives with loading values ≥ |0.15| were considered as the most emblematic of each mixture. This point was only for description purposes, since all factor loadings are displayed in the result tables and all additives contributed to the mixture score calculation. More details on NMF method are provided in eMethod4 in S1 Appendix. Sensitivity analyses using different decomposition algorithms are shown in Table C.b. and c. We checked the stability of food additive mixture intakes over time by performing NMF analysis on two periods of 7.5 years each (corresponding to the median follow-up: period 1 = 2009–2016; period 2 = 2017–2024, in S1 Appendix, Table C.d.). Spearman correlations coefficients were computed to assess the links between NMF components with each other (Table D in S1 Appendix), and between NMF components and food group intakes (Table E in S1 Appendix). Detailed consumption of food groups by

sex-specific quintiles of NMF scores for the mixtures associated with higher type 2 diabetes incidence are provided in Table F in S1 Appendix (added to the initial statistical analysis plan, in response to peer review comments).

The associations between the exposure to food additive mixtures (as reflected by their continuous NMF scores) and higher type 2 diabetes incidence were assessed using multivariable proportional hazard Cox models with age as the time scale. Participants contributed person-time to the models from their age at enrollment in the cohort until their age at the date of type 2 diabetes diagnosis, the date of type 1 diabetes diagnosis, the date of death, the date of last contact, or December 31st 2023, whichever occurred first. Since death and incident type 1 diabetes occurring during follow-up were considered as competing risks for type 2 diabetes, cause-specific Cox models were used. The proportional hazard assumption was checked by examining Schoenfeld residuals and the Grambsch and Therneau's lack-of-fit test (Fig B in S1 Appendix), and the log-linearity of the associations was assessed using restricted cubic splines with three knots at the 27.5th, 50th, and 72.5th percentiles of each mixture score's distribution [22]. Hazard ratios (HRs) and 95% confidence intervals (95% CIs) were computed for a standardized increment of one standard deviation (SD) of each mixture score. Increments are specified in the legend of Fig 1.

The main model was adjusted for a set of predefined risk factors of type 2 diabetes, i.e.,: age (time-scale), sex, Body Mass Index (BMI, continuous, kg/m$^2$), physical activity (categorical IPAQ variable: high, moderate, low), smoking status (never smoked, former smoker, current smokers), number of smoked cigarettes in pack-years (continuous), educational level (did not complete secondary education/up to 2 years of university studies/bachelor degree or higher), socio-professional categories (farmer, craftsman/shopkeeper/entrepreneur, managerial staff/intellectual profession, intermediate profession, employee, manual worker, retired, unemployed, student, and other without professional activity—added following peer review comments), monthly income per household unit (<1,200 €/month; 1,200–1800 €/month; 1800–2,700 €/month; >2,700 €/month—added following peer review comments), family history of type 2 diabetes (yes/no), number of dietary records (continuous), intakes of energy without alcohol (continuous, kcal/d), saturated fatty acids (continuous, g/d), sodium (continuous, mg/d), dietary fiber (continuous, g/d), alcohol (continuous, g/d), and added sugars (continuous, g/d). Sensitivity analyses are presented in eMethods4 and Table G in S1 Appendix, including a model further adjusted for an indicator of health-seeking behaviors and geographical region (coding available in S1 Appendix in footnotes to Table G, Model 6 and 7– added following peer review comments). Exposure coded as tertiles was also tested (Table H in S1 Appendix).

Following peer review comments, we also performed the following set of sensitivity analyses for mixtures 2 and 5: we computed the main model, stratified by an indicator of the nutritional quality of the diet, i.e., the Programme National Nutrition Santé-Guidelines Score 2 (PNNS-G2, below/above the sex-specific median, Table I) We also explored the hypothesis that beyond associations of individual additives with type 2 diabetes incidence, there may be an influences of the mixtures themselves. For this: (1) we adjusted the main Cox model for each food additive characteristic of the mixture, using the residual method (Table J in S1 Appendix); (2) two-by-two interactions between the key food additives contributing to each mixture were formally tested with food additive intakes standardized to mean = 0 and SD = 1 for interpretability (Table K in S1 Appendix). Last, we tested whether the mixtures found associated with type 2 diabetes incidence contributed to mediate the associations between the food groups most associated with these mixtures and incidence of type 2 diabetes risk, using the CMAVERSE R package (Table L in S1 Appendix). All statistical tests were two-sided, and $p$-values < 0.05 were considered statistically significant, except for interaction tests (less powerful), for which 0.1 was considered statistically significant. Analyses were conducted in R version 4.3.3, except for the Restricted Cubic Splines, which were implemented in SAS version

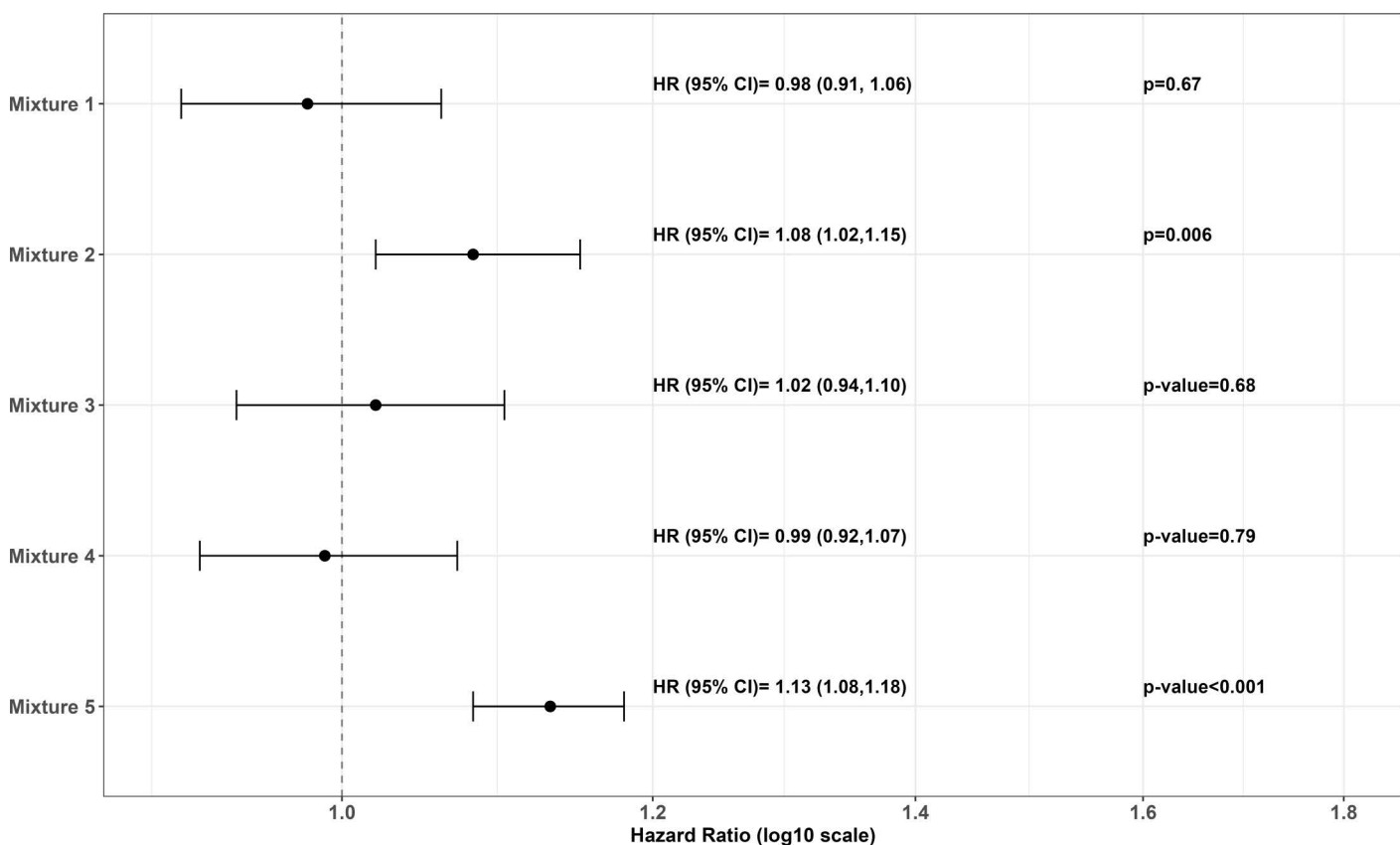

**Fig 1. Associations between food additive mixtures and type 2 diabetes incidence, NutriNet-Santé cohort, 2009–2023 (*n* = 108,643 participants; 1,131 incident cases).** Abbreviations: HR, hazard ratio; CI, confidence interval. Mixtures of food additives were derived from nonnegative matrix factorization (NMF, eMethod4 in S1 Appendix). HRs were computed for increments of 1 standard deviation (SD) of each mixture score: mixture 1 (SD = 12.9), mixture 2 (SD = 8.0), mixture 3 (SD = 88.3), mixture 4 (SD = 20.1), mixture 5 (SD = 14.7). The identified mixtures correspond to food additive combinations derived from the nonnegative matrix factorization (NMF) procedure. Mixture 1 was mainly characterized by sodium carbonates, diphosphates, glycerol, ammonium carbonates, potassium carbonates, and sorbitol. Mixture 2 was mainly characterized by modified starches, pectin, guar gum, carrageenan, polyphosphates, potassium sorbates, curcumin, and xanthan gum. The additives that contributed most to mixture 3 were magnesium carbonates, riboflavin, alpha-tocopherol, and ammonium carbonates. Main contributors to mixture 4 were ammonium carbonates, sodium carbonates, diphosphates, alpha-tocopherol, mono and diacetyl tartaric acid esters of mono and diglycerides of fatty acids, magnesium carbonates, and lecithins. Finally, the main food additives characterizing mixture 5 were citric acid, sodium citrates, phosphoric acid, sulphite ammonia caramel, acesulfame-K, aspartame, sucralose, arabic gum, malic acid, carnauba wax, paprika extract, anthocyanins, guar gum, and pectin. Multivariable Cox proportional hazard models were adjusted for age (time-scale), sex, Body Mass Index (BMI, continuous, kg/m²), physical activity (categorical International Physical Activity Questionnaire (IPAQ) variable: high, moderate, low), smoking status (never smoked, former smoker, current smokers), number of smoked cigarettes in pack-years (continuous), educational level (did not complete secondary education/ up to two years of university studies/ bachelor degree or higher), family history of type 2 diabetes (yes/no), number of dietary records (continuous), socio-professional categories (farmer, craftsman/shopkeeper/ entrepreneur, managerial staff/intellectual profession, intermediate profession, employee, manual worker, retired, unemployed, student, and other without professional activity), monthly household income per consumption unit (<1,200 €/month; 1,200–1,800 €/month; 1,800–2,700 €/month; >2,700 €/month), intakes of energy without alcohol (continuous, kcal/d), saturated fatty acids (continuous, g/d), sodium (continuous, mg/d), dietary fiber (continuous, g/d), alcohol (continuous, g/d), added sugars (continuous, g/d).

9.4. This study is reported as per the Strengthening the Reporting of Observational Studies in Epidemiology (STROBE) guideline (S1 Checklist in STROBE Checklist).

## Results

### Descriptive characteristics

A total of 108,643 participants from the NutriNet-Santé cohort were included in this study (Fig A in S1 Appendix), among which 79.2% were women. At baseline, the median age of the

cohort was 41.2 years (25th–75th percentiles: 29.8–54.5 years). Among the overall cohort, 3.82% ($n = 4,150$) participants have died since their inclusion (2,411 in the present population study) and 9.5% dropped out because they did not want to receive any more questionnaires. Participants included in this study completed a median of 5 dietary records (25th–75th percentiles: 3–9). Their characteristics are detailed in Table 1. UPF (NOVA 4) accounted for a median of 33.8% (25th–75th percentiles: 25.2%–43.7%) of daily energy intake. Daily food additive intakes are described in Table A in S1 Appendix (mean, SD, median, percentage of consumers). A total of 75 food additives were consumed by at least 5% of the participants and were therefore included in NMF mixture analyses.

## Food additive mixtures derived by NMF

The NMF procedure identified five main food additive mixtures (Table B in S1 Appendix: full results). Table 2 summarizes the food additives that were the most emblematic of each mixture. Mixture 1 was mainly characterized by sodium carbonates, diphosphates, glycerol, ammonium carbonates, potassium carbonates, and sorbitol. Mixture 2 was characterized by modified starches, pectin, guar gum, carrageenan, polyphosphates, potassium sorbates, curcumin, and xanthan gum. The additives that contributed most to mixture 3 were magnesium carbonates, riboflavin, alpha-tocopherol, and ammonium carbonates. Main contributors to mixture 4 were ammonium carbonates, sodium carbonates, diphosphates, alpha-tocopherol, mono and diacetyl tartaric acid esters of mono and diglycerides of fatty acids, magnesium carbonates, and lecithins. Finally, the main food additives in mixture 5 were citric acid, sodium citrates, phosphoric acid, sulphite ammonia caramel, acesulfame-K, aspartame, sucralose, arabic gum, malic acid, carnauba wax, paprika extract, anthocyanins, guar gum, and pectin. Mean (SD) NMF scores of the participants for the five mixtures were as follows: mixture 1: 9.1 (12.9); mixture 2: 8.9 (8.0); mixture 3: 9.1 (88.3); mixture 4: 9.1 (20.1); mixture 5: 9.0 (14.7). NMF performed using different algorithms (Table C.b. and C.c. in S1 Appendix) and computed at the two periods (corresponding to the first and second halves of the cohort follow-up, Table C.d. in S1 Appendix) indicated overall robustness of the findings and stability of the mixtures over time.

Overall, there was little correlation between the five mixtures, the highest being observed between mixtures 1 and 4: Spearman correlation coefficient $\rho = 0.39$ (Table D in S1 Appendix). Table E in S1 Appendix displays the correlations between additive mixtures and food group intakes. Detailed consumption data for specific food groups, stratified by sex-specific quintiles of participants, are additionally provided for mixtures 2 and 5 (Table F in S1 Appendix). Mixture 1 was correlated with cakes and biscuits ($\rho = 0.35$) as well as savory snacks ($\rho = 0.18$). The food groups most correlated with mixture 2 were broth ($\rho = 0.40$), dairy desserts ($\rho = 0.22$), and fats and sauces ($\rho = 0.21$). No specific food group as a whole correlated with mixture 3. Indeed, the additives contained in this mixture are used in foods that are frequently consumed, but distributed in an isolated way in multiple food groups (e.g., E504 magnesium carbonate in table-top salt, in certain brands of energy drinks, as well as in specific brands of chocolate cookies and cocoa powder, etc.). In line with the correlation between mixtures 1 and 4, mixture 4 was also correlated with savory snacks ($\rho = 0.19$) and cakes and biscuits ($\rho = 0.18$). The food groups most correlated with mixture 5 were artificially sweetened soft drinks ($\rho = 0.41$) and sugary drinks ($\rho = 0.37$).

## Associations between food additive mixtures and type 2 diabetes incidence

A total of 1,131 incident type 2 diabetes were detected (mean follow-up duration = 7.7 years (SD 4.6)). Schoenfeld residuals (Fig B in S1 Appendix) did not show evidence for violation of

**Table 1. Baseline characteristics of study participants, NutriNet-Santé cohort, 2009–2023 (N = 108,643).**

|  | Mean (SD) or N (%) | Median [25th–75th percentiles] |
|---|---|---|
| Age (years) | 42.5 (14.6) | 41.2 [29.8, 54.5] |
| Sex, Female | 86032 (79.2%) |  |
| BMI (kg/m²)* | 23.6 (4.4) | 22.7 [20.6, 25.5] |
| Family history of type 2 diabetes[a],* | 16994 (15.6%) |  |
| IPAQ physical activity level* |  |  |
| High | 30928 (28.5%) | .. |
| Moderate | 40415 (37.2%) | .. |
| Low | 22535 (20.7%%) | .. |
| Smoking status* |  |  |
| Never | 54613 (50.3%) | .. |
| Former smoker | 35205 (32.4%) | .. |
| Current | 18502 (17.0%) | .. |
| Educational level* |  |  |
| Did not complete secondary education | 18997 (17.5%) | .. |
| Up to two years of university studies | 52740 (48.5%) | .. |
| Bachelor degree or higher | 35863 (33.0%) | .. |
| Socio-professional categories |  |  |
| Farmer | 333 (0.3%) | .. |
| Craftsman/shopkeeper/entrepreneur | 1975 (1.8%) | .. |
| Managerial staff/intellectual profession | 25527 (23.5%) | .. |
| Intermediate profession | 18321 (16.9%) | .. |
| Employee | 20158 (18.5%) | .. |
| Manual worker | 1397 (1.3%) | .. |
| Retired | 17473 (16.1%) | .. |
| Unemployed | 6337 (5.8%) | .. |
| Student | 6756 (6.2%) | .. |
| Other without professional activity | 9956 (9.2%) | .. |
| Monthly household income per consumption unit (euros) |  |  |
| <1200 | 18878 (17.4%) | .. |
| 1200–1800 | 26327 (24.2%) | .. |
| 1800–2700 | 24900 (23.0%) | .. |
| >2700 | 25365 (23.3%) | .. |
| Geographical region |  |  |
| North | 11517 (10.6%) | .. |
| North-East | 12805 (11.8%) | .. |
| West | 13226 (12.2%) | .. |
| Center | 4104 (3.8%) | .. |
| South-West | 19660 (18.1%) | .. |
| South-East | 23026 (21.2%) | .. |
| Ile-deFrance | 20954 (19.3%) | .. |
| French overseas territories and departments | 1024 (0.9%) | .. |
| Corsica | 324 (0.3%) | .. |
| Other | 1992 (1.8%) | .. |
| Energy intake without alcohol (kcal/day) | 1840 (450) | 1790 [1538, 2093] |
| Alcohol intake (g/day) | 7.70 (11.7) | 3.21 [0, 10.6] |
| Saturated fat intake (g/day) | 33.2 (12.1) | 31.9 [64.6, 95.5] |
| Sugar intake (g/day) | 198 (57.4) | 192 [159.6, 229.8] |

*(Continued)*

**Table 1.** (Continued)

|  | Mean (SD) or *N* (%) | Median [25th–75th percentiles] |
|---|---|---|
| Sodium (mg/day) | 2720 (886) | 2600 [2118, 3186] |
| Fiber intake (g/day) | 19.6 (7.3) | 18.5 [14.7, 23.2] |
| Fruits and vegetables intake (g/day) | 464 (231) | 440 [307.2, 590.1] |
| Red and processed meat intake (g/day) | 75.6 (52.6) | 68.8 [38.9, 103.6] |
| Total food additive intake (mg/day) | 5730 (4400) | 4770 [2791, 7510] |

ᵃIn first-degree relatives.

*Missing values: BMI *n* = 3,000; Family history of type 2 diabetes *n* = 333; IPAQ physical activity level *n* = 14,765; Smoking status *n* = 323; Educational level *n* = 1,043; Socio-professional categories *n* = 410; Monthly household income per consumption unit *n* = 13,173, geographical region *n* = 11.

Abbreviations: SD, standard deviation, *N*, number, IPAQ, International Physical Activity Questionnaire, BMI, body mass index.

the proportional hazard assumptions. Associations between food additive mixtures and type 2 diabetes incidence are outlined in Fig A in S1 Appendix. Mixture 2 (HR$_{\text{per an increment of 1SD}}$ = 1.08 [1.02,1.15], *p* = 0.006) and mixture 5 (HR$_{\text{per increment of 1SD}}$ = 1.13 [1.08,1.18], *p* < 0.001) were positively associated with higher type 2 diabetes incidence. No association with type 2 diabetes incidence was observed for mixtures 1, 3, and 4. Sensitivity analyses fully aligned with results from the main model, supporting the robustness of the findings (Table G in S1 Appendix). These models tested several further or modified adjustments (for prevalent metabolic disorders; for other mixtures - mutual adjustment; for Healthy and Western dietary patterns; for food groups instead of nutrients; for health-seeking behaviors; and for geographical region), and exclusion of cases diagnosed in the first 2 years of follow-up to challenge reverse causality. Restricted cubic spline plots confirmed the linearity of the observed associations for all mixtures except mixture 3 (Fig 2 for mixtures 2 and 5 and Fig C for all in S1 Appendix). Categorical analyses (tertiles) were also conducted and showed similar results (Table H in S1 Appendix).

For mixtures 2 and 5: very similar associations were observed in both PNNS-GS2 strata (although borderline nonsignificant for mixture 2, likely due to halved statistical power) and no interaction was detected between the PNNS-GS2 score and the additive mixtures (*p* for interaction = 0.8 for mixture 2 and 0.9 for mixture 5), supporting an association between these mixtures and type 2 diabetes incidence, independent from the nutritional quality of the diet (Table I in S1 Appendix). Despite slight attenuations, the associations between mixtures 2 and 5 and type 2 diabetes incidence remained similar after adjustment for each food additive characteristic of the mixture using the residual method, suggesting that the associations were not strongly driven by a unique additive alone (Table J in S1 Appendix). Out of the 28 two-by-two interactions tested between the 8 additives most emblematic of mixture 2, 3 were detected (i.e., *p* for interaction < 0.1) with beta coefficient >0 (suggesting synergism), and 4 were detected with beta coefficient <0 (suggesting antagonism). For mixture 5, 91 interactions were tested between the 14 most emblematic additives: 6 were detected with beta coefficient >0 and 4 with beta coefficient <0 (Table K in S1 Appendix). Lastly, mediation analyses were conducted (Table L in S1 Appendix). Among food groups most correlated with additives mixtures, fats and sauces (correlated with mixture 2), artificially sweetened beverages, and sugary drinks (both correlated with mixture 5) were associated with higher type 2 diabetes incidence (Table L in S1 Appendix, Cox models). The mediation of the association between fats and sauces and type 2 diabetes by mixture 2 was modest (proportion of the association mediated = 18%, *p*-value = 0.09). Mixture 5 mediated 42% of the association between sugary

**Table 2. Food additive mixtures identified by nonnegative matrix factorization: loading values of main additive contributors[a], NutriNet-Santé cohort, 2009–2023[b].**

| Mixture 1 Food additive/Loading value | | Mixture 2 Food additive/Loading value | | Mixture 3 Food additive/Loading value | | Mixture 4 Food additive/Loading value | | Mixture 5 Food additive/Loading value | |
|---|---|---|---|---|---|---|---|---|---|
| E500 Sodium carbonates | 0.99 | Modified starches | 0.99 | E504 Magnesium carbonates | 0.99 | E503 Ammonium carbonates | 0.99 | E330 Citric acid | 0.83 |
| E450 Diphosphates | 0.78 | E440 Pectins | 0.31 | E101 Riboflavin | 0.53 | E500 Sodium carbonates | 0.35 | E331 Sodium citrates | 0.63 |
| E422 Glycerol | 0.37 | E412 Guar gum | 0.26 | E307 Alpha-tocopherol | 0.24 | E450 Diphosphates | 0.30 | E338 Phosphoric acid | 0.59 |
| E503 Ammonium carbonates | 0.35 | E407 Carrageenan | 0.24 | E503 Ammonium carbonates | 0.17 | E307 Alpha-tocopherol | 0.28 | E150d Sulphite ammonia caramel | 0.59 |
| E501 Potassium carbonates | 0.17 | E452 Polyphosphates | 0.21 | | | E472e DATEM[c] | 0.18 | E950 Acesulfame K | 0.56 |
| E420 Sorbitols | 0.16 | E202 Potassium sorbate | 0.17 | | | E504 Magnesium carbonates | 0.17 | E951 Aspartame | 0.41 |
| | | E100 Curcumin | 0.16 | | | E322 Lecithins | 0.15 | E955 Sucralose | 0.25 |
| | | E415 Xanthan gum | 0.16 | | | | | E414 Arabic gum | 0.23 |
| | | | | | | | | E296 Malic acid | 0.19 |
| | | | | | | | | E903 Carnauba wax | 0.18 |
| | | | | | | | | E160c Paprika extract, capsanthin, capsorubin | 0.17 |
| | | | | | | | | E163 Anthocyanins | 0.15 |
| | | | | | | | | E412 Guar gum | 0.15 |
| | | | | | | | | E440 Pectins | 0.15 |

[a]The loading values indicate the strength of the association between the specific food additive and each NMF mixture score.

[b]For conciseness purposes, only food additives with NMF loading values ≥│0.15│ are displayed here. Full results are presented in Table C in S1 Appendix.

[c]Mono and diacetyl tartaric acid esters of mono and diglycerides of fatty acid.

drinks and type 2 diabetes ($p < 0.001$) and 52% of the association between artificially sweetened beverages and type 2 diabetes ($p = 0.03$).

## Discussion

This study identified a positive association between two broadly ingested food additive mixtures and a higher incidence of type 2 diabetes. One of these mixtures (mixture 2) was primarily composed of several emulsifiers (modified starches; pectin; guar gum; carrageenan; polyphosphates; xanthan gum), in addition to a preservative (potassium sorbate), and a dye (curcumin), which are typically found in a variety of industrially-processed foods, including broth, dairy desserts, fats and sauces. The other mixture associated with type 2 diabetes incidence (mixture 5) was primarily composed of food additives found in artificially sweetened beverages and sugary drinks. These additives included acidifiers and acid regulators (citric acid; sodium citrates; phosphoric acid; malic acid), dyes (sulphite ammonia caramel, which is characteristic of cola sodas; anthocyanins; paprika extract), artificial sweeteners (acesulfame-K; aspartame; sucralose), and some emulsifiers (arabic gum; pectin; guar gum). Exploratory analyses suggested both synergistic and antagonist interactions between several food additives emblematic of these mixtures. In order to account for the nutritional quality of the diet and isolate as much as possible potential effects from food additives, all models were systematically adjusted for key nutritional intakes and no interaction was observed with

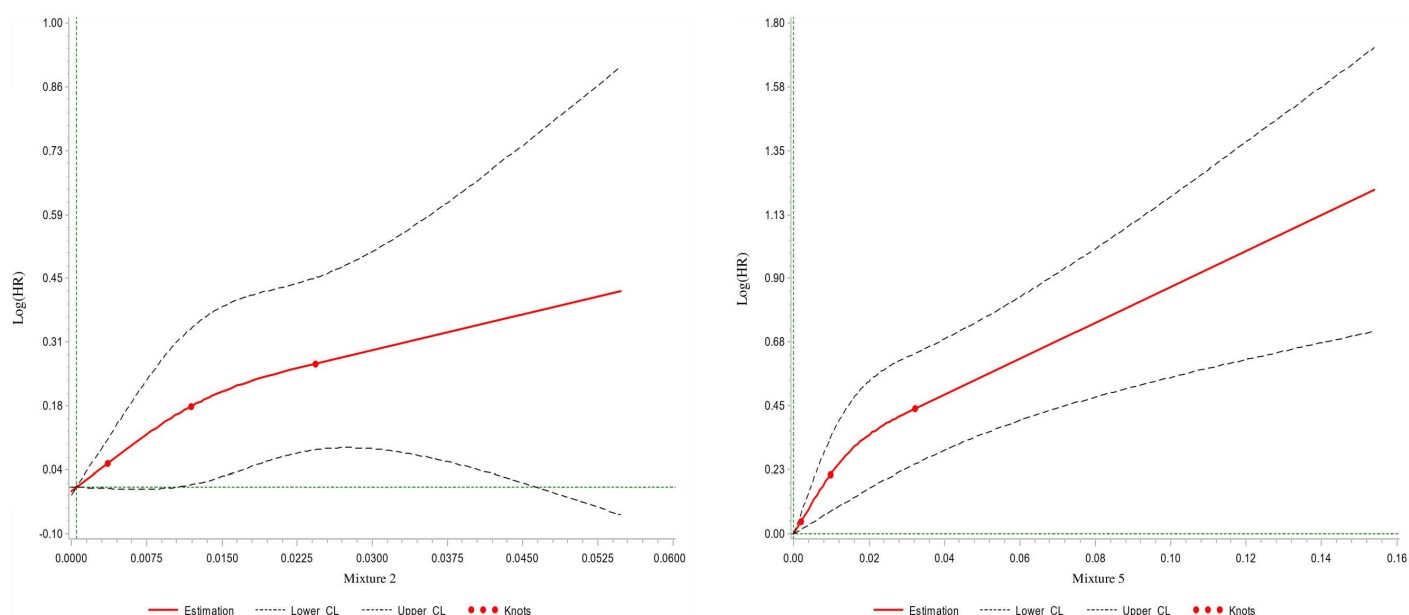

**Fig 2. Dose-response associations between food additive mixtures 2 and 5 and type 2 diabetes incidence, restricted cubic spline plots, NutriNet-Santé cohort, 2009-2023 ($n$ = 108,643 participants; 1,131 incident cases).** Abbreviations: CL confidence limit. Mixture 2: P-value for nonlinearity = 0.4; Mixture 5: *P*-value for nonlinearity = 0.05. Mixture 2 was mainly characterized by modified starches, pectin, guar gum, carrageenan, polyphosphates, potassium sorbates, curcumin, and xanthan gum. Mixture 5 was mainly characterized by citric acid, sodium citrates, phosphoric acid, sulphite ammonia caramel, acesulfame-K, aspartame, sucralose, arabic gum, malic acid, carnauba wax, paprika extract, anthocyanins, guar gum, and pectin.

an overall indicator of the nutritional quality of the diet. The associations between additive mixtures and type 2 diabetes incidence were therefore studied 'all other things being equal' in terms of intakes of sugar, saturated fat, energy, alcohol, and so forth.

To our knowledge, this study is the first to evaluate and detect positive associations between food additive mixtures and higher type 2 diabetes incidence in a large prospective cohort. Thus, direct comparison of our findings with previous epidemiological literature is not possible. However, our results can be put into perspective with those of previous epidemiological studies that have examined associations between single food additives and the incidence of type 2 diabetes, which, to our knowledge, are only available in the NutriNet-Santé cohort to date. Consistently, several of the food additives emblematic of mixtures 2 or 5 were associated with higher type 2 diabetes incidence in previous publications on emulsifiers and artificial sweeteners [5,6] (preservatives and dyes are currently under investigation): carrageenans ($HR_{per\ increment\ of\ 100\ mg\ per\ day}$ = 1.03 [1.01,1.05], $p < 0.0001$), sodium citrate ($HR_{per\ increment\ of\ 500\ mg\ per\ day}$ = 1.04 [1.01,1.07], $p = 0.0080$), guar gum ($HR_{per\ increment\ of\ 500\ mg\ per\ day}$ = 1.11 [1.06,1.17], $p < 0.0001$), gum arabic ($HR_{per\ increment\ of\ 1,000\ mg\ per\ day}$ = 1.03 [1.01,1.05], $p = 0.013$), xanthan gum ($HR_{per\ increment\ of\ 500\ mg\ per\ day}$ = 1.08 [1.02,1.14], $p = 0.013$), aspartame ($HR_{for\ an\ increment\ of\ 100\ mg/day}$ = 1.26 (1.08,1.46), $p = 0.003$), acesulfame-K ($HR_{for\ an\ increment\ of\ 100\ mg/day}$ = 1.62 (1.12,2.33), $p = 0.010$).

In these previous papers published on food additives and type 2 diabetes incidence in NutriNet-Santé [5,6,23], the objective was to investigate one specific food additive (or food additive category), independently from the intake of other additives. Therefore, the models (main or sensitivity) were adjusted for the proportion of UPF in the diet (as an overall indicator of multiple additive exposure), and/or for other additives than the one studied (e.g., intake of all emulsifiers except the one studied). These analyses suggested potential effects of these additives (some artificial sweeteners, emulsifiers, nitrites) per se, consistent with in vivo/in vitro experimental/mechanistic data—see below [24–26]. In the latter, the effect of each

specific additive is isolated and tested, thus, results cannot be attributed to a potential mixture effect. The present study is complementary. Our hypothesis was that beyond potential influence of individual additives, the mixtures themselves may play a role, resulting from interactions between food additives. Our results suggest that the associations between mixtures 2 and 5 and type 2 diabetes incidence were not entirely driven by any of the specific additives alone. Besides, several interactions have been detected between emblematic additives of mixtures 2 and 5. However, the number of detected interactions was limited compared to the overall number tested. Moreover, both synergistic, but also antagonist interactions were observed. The relative influence of individual additives versus their interactions should be explored in future mechanistic investigations. To our knowledge, only one human study previously explored the potential health impact of a food additive mixture [27]: a randomized control trial conducted in the UK observed that a 6-week exposure to a pre-defined dye mixture (i.e., carmoisine [E122], sunset yellow [E110], tartrazine [E102], ponceau 4R [E124], allura red AC [E129]; 20 or 30 mg for the 3-year-old children and 24.98 or 62.4 mg for 8/9-year-old children) and 45 mg of sodium benzoate preservative [E211] increased hyperactivity among 3-year-old and 8/9-year-old children.

Our results are supported by several experimental in vivo and in vitro studies that suggest deleterious effects for several food additives emblematic of mixtures 2 and 5 and thus could explain the associations with higher type 2 diabetes incidence found in this study. For example, guar gum [E412], present both in mixtures 2 and 5, has been pointed out for its alteration of gut microbiota composition leading to an elevation of pro-inflammatory markers and potential metabolic perturbations in a mouse model [28]. Evidence of the role of the gut microbiota's contribution in type 2 diabetes mellitus is growing, especially through the alteration of glucose metabolism pathways [29]. Carrageenan [E407], an emulsifier predominant in mixture 2, was observed to impair glucose metabolism in mice and showed inflammatory properties [30,31], which may be involved in type 2 diabetes etiology. The role of artificial sweeteners such as the ones found in mixture 5 on gut microbiota perturbations has also been suggested [32]. In particular, acesulfame-K [E950] and sucralose [E955] were observed to shape the microbial populations and lead to dysbiosis [33,34], which in turn may enhance glucose intolerance and changes in host physiology in mice. This was supported by the observation of similarities in microbial populations from noncaloric artificial sweeteners consumers and patients with type 2 diabetes [35]. Besides, some experimental results suggest that different additives may interact and thus lead to synergistic or antagonist effects [27,36]. In particular, an in vitro study assessed the neurotoxicity of two food additive 'mixtures (i.e., Brilliant Blue [E133] and ʟ-glutamic acid [E621], or Quinoline Yellow [E104] and aspartame [E951]) and reported that their synergy was potentially more toxic than the effect of the individual compounds [37]. Behavior and brain changes have been reported in rats when exposed to food-colorant mixtures [38]. In an in vitro experiment, Meng and colleagues observed cytotoxic effects caused by multiple food additives (Sodium benzoate, Potassium sorbate, New red, Caffeine, Sodium saccharin, Acesulfame K, Aspartame, and Tartrazine, Sunset Yellow, Erythrosine, Amaranth, Ponceau 4R, Brilliant blue FCF, Allura red AC) at doses below the tolerable upper levels in beverages [9]. In a recent paper based on four human cell models, we observed cyto-/genotoxic effects of some additive mixtures beyond the ones observed for the substances alone [10]. Further experimental studies are needed to gain a deeper understanding of potential interaction (synergistic and antagonist) effects of food additive mixtures on metabolism and diabetes etiology.

The strengths of this study lie in its prospective design, large sample size, meticulous assessment of dietary intakes, and unique data on the exposure to a broad range of food additives. The NutriNet-Santé study stands out as the first, to our knowledge, to evaluate both

qualitative and quantitative exposures to food additives, leveraging detailed brand-specific and repeated 24-h dietary records, links to multiple food composition databases (Observatoire de la Qualité de l'Alimentation [OQALI], Open Food Facts, Global New Products Database [GNPD], EFSA, and General Standard for Food Additives [GSFA]), ad-hoc laboratory assays, and dynamic matching to account for reformulations of industrial food items over time [8]. The studied mixtures are clinically relevant because they represent the ones to which consumers are most frequently exposed daily. Indeed, the approach was to derive NMF components based on the observation of real-life consumption data in this large-scale population study. Besides, associations remained stable across various sensitivity analyses and are supported by mechanistic plausibility.

Several limitations should be acknowledged. First, the observational nature of the design introduces inherent constraints and a single observational epidemiological study is not sufficient per se to establish causality. Despite extensive adjustments for confounding variables, including dietary, lifestyle, anthropometric, and socio-demographic factors, the potential for unmeasured and residual confounding persists. Race/ethnicity and religion were not available in the cohort due to a very restrictive ethical/legal regulation policy regarding the collection of these data in French epidemiological studies (specific authorizations needed). Second, exposure to food additives has not been validated against blood or urine assays due to lack of existing specific biomarkers so far. Classification errors on exposure (e.g., in products exempt from labeling requirements) or covariates could not be entirely ruled out. However, numerous methodological studies in e-epidemiology were conducted and published, to challenge the reliability of the information collected online in this cohort. These included comparisons with traditional data collection methods (e.g., paper-and-pencil or interviews by trained dietitians) and against gold standards, such as blood and urine biomarkers of nutritional intakes and investigator-measured height and weight [12,14–16,39,40]. High consistency was observed between online tool and standard methods, supporting the reliability of the data collected. The methodological studies even highlighted the advantage of online web questionnaires in increasing the quality of collected data and reducing input error and outliers thanks to integrated field controls and automated conditional skips. Several studies also showed that web-based data collection may even increase accuracy by lowering social desirability bias associated with face-to-face interviews [41–43]. It is merely impossible to guaranty a perfect measurement of exposure and covariates in any population-based epidemiological study in real-life setting. However, in etiological studies, the ability to rank participants with contrasted exposures between themselves is more important than reaching perfectly accurate absolute values. Besides, misclassification in exposure or covariates may have induced a nondifferential measurement error (identically in future cases and noncases given the prospective design), but although an overestimation cannot be excluded, it most probably led to an underestimation of the observed associations. Third, several food additives were ingested by a very low number of individuals and thus, could not be included in NMF mixture analyses. In addition, food additive intake may vary over time. Yet, NMF analyses performed at the two time periods showed an overall high stability of the five main mixtures across follow-up. Next, despite the multi-source case ascertainment strategy (which combined self-reported disease and medication use, linkage to a national health insurance database, and fasting blood glucose measurements in a subsample), under-detection of certain undiagnosed type 2 diabetes cases could not be entirely ruled out. In France, the prevalence of undiagnosed diabetes cases is estimated around 1.7% (IC95% 1.1–2.4), higher in men (2.7% IC95% 1.4–4.0) than in women (0.9% IC95% 0.3–1.4) [44]. Last, the generalizability of our findings may be influenced by the cohort's demographic characteristics, such as a higher proportion of women and a more health-conscious population. Therefore, caution is warranted when extrapolating our results to broader populations.

In conclusion, the results of this large population-based study revealed positive associations between widely consumed food additive mixtures and a higher incidence of type 2 diabetes. To our knowledge, these findings provide the first insight into the food additives that are frequently ingested together [due to their co-occurrence in industrially-processed food products or resulting from the co-ingestion of foods in dietary patterns) and how these additive mixtures may be involved in type 2 diabetes etiology. Further long-term observational epidemiological studies, as well as short-term interventions and pre-clinical experimental research are required to elucidate the underlying mechanisms and gain a deeper understanding of potential synergies and antagonisms between these food chemicals. These results suggest that it may be of interest to consider potential interaction/synergistic/antagonist effects when assessing the safety of food additives and call for a reevaluation of regulations governing their use by the food industry, with the aim of enhancing consumer protection. In the meantime, these findings provide support for the public health recommendation to limit exposure to UPF and their nonessential food additives.

## Supporting information

**S1 Checklist. STROBE checklist.**
(DOC)

**S1 Appendix. eMethods. eResults. Fig A.** Flowchart, NutriNet-Santé cohort, 2009-2023. **Table A.** Daily food additive intakes (mg/d) among study participants from the NutriNet-Santé cohort, 2009–2023 (*N* = 108,643)[a,b]. **Table B.** Food additive mixtures identified by nonnegative matrix factorization[a]. **Table C.** Consensus map for rank number determination and sensitivity analyses using other decomposition algorithms in the NMF procedure. **Table D.** Spearman correlations between the five NMF food additive mixtures. **Table E.** Spearman correlations between NMF food additive mixtures and food group intakes. **Table F.** Food group consumption of participants according to sex-specific quintiles of mixtures 2 and 5[a]. **Fig B.** Correlations between Schoenfeld residuals and timescale (age, y) from multivariable Cox models between food additive mixtures and type 2 diabetes incidence, NutriNet-Santé cohort, 2009–2023 (*n* = 108,643). **Fig C.** Dose-response associations between food additive mixtures and type 2 diabetes incidence, restricted cubic spline plots, NutriNet-Santé cohort, 2009–2023 (*n* = 108,643 participants; 1,131 incident cases). **Table G.** Associations between food additive mixtures and type 2 diabetes incidence, NutriNet-Santé cohort, 2009–2023—Sensitivity analyses. **Table H.** Association between food additive mixtures coded as tertiles and incidence of type 2 diabetes, NutriNet-Santé cohort, 2009–2023—Sensitivity analyses. **Table I.** Associations between food additive mixtures and incidence of type 2 diabetes, stratified by the Programme National Nutrition Santé-Guidelines Score 2 (PNNS-GS2), NutriNet-Santé cohort, 2009-2023—Sensitivity analyses. **Table J.** Association between mixtures 2 and 5 and type 2 diabetes incidence adjusted for the key food additives contributing to each mixture (residual method), NutriNet-Santé cohort, 2009–2023 (*n* = 108,643 participants; 1,131 incident cases) —Sensitivity analyses. **Table K.** Interactions between the key food additives contributing to mixtures 2 and 5, NutriNet-Santé cohort, 2009–2023 (*n* = 108,643 participants; 1,131 incident cases). **Table L.** Mediation analyses.
(DOCX)

## Acknowledgments

We thank Thi Hong Van Duong, Régis Gatibelza, Jagatjit Mohinder, and Aladi Timera (computer scientists), Selim Aloui (IT manager); Julien Allègre, Nathalie Arnault (data-manager/statisticians); Paola Yvroud (health event validator); Maria Gomes and Mirette

Foham (participant support); Marine Ricau (Operational coordinator); Marie Ajanohun, Souad Hadji (administration and finance), and Nadia Khemache (Administrative manager), for their technical contribution to the NutriNet-Santé study. We also warmly thank all the volunteers of the NutriNet-Santé cohort.

*IARC disclaimer:* Where authors are identified as personnel of the International Agency for Research on Cancer/World Health Organization, the authors alone are responsible for the views expressed in this article and they do not necessarily represent the decisions, policy or views of the International Agency for Research on Cancer/World Health Organization.

## Author contributions

**Conceptualization:** Marie Payen de la Garanderie, Benoît Chassaing, Mathilde Touvier.

**Data curation:** Fabien Szabo de Edelenyi, Laurent Bourhis.

**Formal analysis:** Marie Payen de la Garanderie, Nicolas Deschamp, Mathilde Touvier.

**Funding acquisition:** Mathilde Touvier.

**Investigation:** Marie Payen de la Garanderie, Mathilde Touvier.

**Methodology:** Raphaël Porcher, Mathilde Touvier.

**Project administration:** Mathilde Touvier.

**Resources:** Cédric Agaësse, Alexandre De Sa, Raphaël Porcher, Mathilde Touvier.

**Supervision:** Nicolas Deschamp, Guillaume Javaux, Mathilde Touvier.

**Validation:** Mathilde Touvier.

**Writing – original draft:** Marie Payen de la Garanderie.

**Writing – review & editing:** Anaïs Hasenbohler, Fabrice Pierre, Xavier Coumoul, Emmanuelle Kesse-Guyot, Benjamin Allès, Léopold K. Fezeu, Emmanuel Cosson, Sopio Tatulashvili, Inge Huybrechts, Serge Hercberg, Mélanie Deschasaux-Tanguy, Benoît Chassaing, Héloïse Rytter, Bernard Srour, Mathilde Touvier.

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
