## [Editor Report · Decision Letter 0]

12 Sep 2024

Dear Dr Payen de la Garanderie, 

Thank you for submitting your manuscript entitled "Food additive mixtures and risk of type 2 diabetes: results from the NutriNet-Santé cohort" for consideration by PLOS Medicine.

Your manuscript has now been evaluated by the PLOS Medicine editorial staff and I am writing to let you know that we would like to send your submission out for external peer review.

Please re-submit your manuscript within two working days, i.e. by Sep 16 2024.

Feel free to email me at atosun@plos.org or us at plosmedicine@plos.org if you have any queries relating to your submission.

Kind regards,

Alexandra Tosun, PhD

Associate Editor

PLOS Medicine

---

## [Decision Letter · Decision Letter 1]

25 Oct 2024

Dear Dr Payen de la Garanderie,

Many thanks for submitting your manuscript "Food additive mixtures and risk of type 2 diabetes: results from the NutriNet-Santé cohort" (PMEDICINE-D-24-03030R1) to PLOS Medicine. The paper has been reviewed by subject experts and a statistician; their comments are included below and can also be accessed here: [LINK]

As you will see, the reviewers found the study to be interesting, but raised several points for clarification. After discussing the paper with the editorial team and an academic editor with relevant expertise, I'm pleased to invite you to revise the paper in response to the reviewers' comments. Please note that the manuscript has generated extensive discussion among the team, so we would like to emphasize that the reviewers' and academic editor's comments should be carefully considered. We plan to send the revised paper to some or all of the original reviewers, and we cannot provide any guarantees at this stage regarding publication.

We ask that you submit your revision by Nov 15 2024. However, if this deadline is not feasible, please contact me by email, and we can discuss a suitable alternative.

Don't hesitate to contact me directly with any questions (atosun@plos.org). 

Best regards, 

Alexandra 

Alexandra Tosun, PhD 

Associate Editor

PLOS Medicine

atosun@plos.org

Comments from the academic editor:

The authors conducted an observational study relating 'mixtures' of food additives to incidence of type 2 diabetes.

The major and minor comments are supplied hereafter.

Major comments:

0. Some of the individual food additives were evaluated previously, as cited #5 (Lancet Diabetes Endocrinol, 2024) and were shown to be associated with type 2 diabetes incidence in the same population. Because of the associations, the novelty of this current study is weak. Also, two analytic studies from the same cohort could question whether single chemicals or a combination played a role in showing the positive association. The authors discussed it but did not address the question.

The previous study reported in Lancet D&E 2024 had the weakness of mutual confounding between food additives (emulsifiers), and then this study demonstrated it. It would be an important notion, but it remained unclear if the positive association was due to a single factor or a combination. Without such an in-depth investigation, this study seems weak.

The authors argued the "cocktail effect", but a specific analysis to test the presence is absent, and the method is too weak to demonstrate or argue the effect. The authors cannot rule out the possibility that a single agent drove the positive association. Even if the "cocktail effect" looked significant, measurement errors and confounding would remain concerning in the multivariable setting.

The authors should design their analyses to test the effect of multiple food additives above and beyond the individual ones. Of note, several food additives in the statistically identified "mixture" were confirmed to be associated with type 2 diabetes, so the confounded interpretations should be avoided. The authors seem aware of the research question (Page 14 in the Discussion) but not aware of proper methods, but any approaches need to be conducted with or without limitations/assumptions. At least, the authors should cite the Lancet D&E paper and re-justify this study to test the hypothesis of the interaction.

The "Cocktail" effect is not a technical term, and the authors should avoid it. What the authors examined must align with a hypothesis about an interaction. The authors should document their interest using such a known term.

1. The authors documented the limitation of residual confounding, but the effort to reduce it and the highlight seem insufficient. The authors collected the majority of the information through the Internet. Self-reports partly affected by health consciousness must cause differential measurement errors, such as those of body weight and height, as is often the case, particularly among women. Such uncharacterised errors could happen to many covariates, cause insufficient adjustment for those and cause further residual confounding.

When the authors described the study limitations, they argued, "Despite extensive adjustments for confounding variables, including dietary, lifestyle, anthropometric, and socio-demographic factors, ..."

This statement indicated that the authors had identified many covariates to adjust for in their analysis. However, socioeconomic status (SES) was not adjusted for well. It likely contributed to dietary behaviours, health consciousness, health-seeking behaviours, and others that influence health outcomes through numerous mechanisms. Despite the well-known confounding roles of SES, the authors included only a single three-level covariate of education history. Just three levels were unlikely to capture the SES. The authors must be aware of the diversity of the study population, including race/ethnic, regional, cultural, and economic diversity across France. The authors did not account for any of those, except for educational attainment. The authors should make a greater effort.

The authors' group may not have considered it in previous studies from the same cohort, but that would not matter. The authors should adjust for a reasonable amount of SES variables, such as household or individual incomes, occupations, race/ethnic status, and religion. (Otherwise, collider bias happens in the current analysis, causing unmeasured confounding.).

Health consciousness is also a potential confounder, as the authors noted. The authors should adjust for participation in clinical screening, access to dental care, and others.

Those covariates would matter in this study because the authors have identified weak positive associations. Weak positive confounders could elevate the likelihood of detecting false-positive associations.

2. 

The authors investigated a linear combination of multiple food additives. However, this does not mean that the authors examined a synergistic or cocktail effect of the multiple food additives. Possibly, only a single food additive within a mixture was causal, whereas other additives had no effect. Taking the possibility, the authors should limit and tone down their conclusion as if there were a cocktail effect. Then, they should argue the need for experimental studies to identify whether only a single causal agent may play a role predominantly or a synergism may happen.

3.

The authors identified type 2 diabetes cases with both subjective (self-reports) and objective information. The approach is great but could miss undiagnosed type 2 diabetes, which could be present to an unignorable degree. Individuals with unrecognized diabetes cannot be identified by self-reports or the national registry. The authors should discuss it.

4.

For each of the "mixture" #2 and #5, the authors should fit a regression model including individual food additives that contribute to the mixture and the mixture itself. If the positive association of the mixture with T2D remains, that should be considered as evidence that the combination matters beyond individual effects.

Similarly, the authors should demonstrate whether or not the positive association was driven by single factor or multiple factors.

Basically, the authors should not give up differentiating between the potential of individual effects and the potential of the effect of the combination or interaction.

These analyses are essential for this manuscript, in addition to the maximal effort to adjust for socioeconomic status and health-seeking behaviours. For the latter confounding, the authors should identify if the participants joined any screening activities, and I hope those are available in their registration data / linked medical records.

5.

In Discussion, the authors should discuss whether the previous studies on emulsifiers and artificial sweeteners should be interpreted. It seems that mutual confounding between food additives substantially matter. The authors should explicitly state that previous findings should be interpreted more than published, with regards to the issue of mutual confounding.

Minor comments:

Risk and incidence should be distinguished.

The aggregates of the food additives are unclear in terms of their clinical relevance. The analyses conducted for the current eTable 4 should be elaborated more to identify food sources and dietary patterns. SES may better be carefully considered, too.

On page 6 and the other pages, the authors should not use "validated" to describe the validity of their measurements. No variable must have been measured with perfection. The authors should document how valid each measure was as much as possible. Given the aim of this work, the authors must be mindful of whether or not each measured variable could identify a between-individual difference rather than its absolute levels of exposure. Thus, the authors should document the quantitative abilities of the exposures and covariates (such as BMI) to rank individuals.

As mentioned above, the measurement error of BMI and behavioural factors were more concerning than usual because the authors recruited individuals over the Internet and collected self-reports of many variables.

Page 8:

The authors documented specifications in their dimension-reduction technique. Those selections are subject to sensitivity analyses.

Comments from the reviewers: 

Reviewer #1: See attachment

Michael Dewey

Reviewer #2: The paper by de la Garanderie et al. Reports a very fancy prospective cohort analysis on the relationship between the consumption of 5 different food additive mixtures and type 2 diabetes incidence, conducted in the large NutriNetSanté cohort in France. 

Investigators first characterized presence/absence and doses of several food additives by leveraging food records completed by participants, ingredient lists coupled with different databases on food additives. This method was previously described and published in Sci Rep.

They then used a data-driven approach (NMF) to identify 5 food additive mixtures. These mixtures represent "clusters" of food additive commonly consumed together due to concomitant presence in the same food for instance. 

They then assessed using cox models the relationship between each mixture and T2D incidence. Two mixtures were found to be positively associated with T2D risk. The first was mostly composed of emulsifiers and the second was composed of food additives commonly present in SSB and ASB. 

Analyses are comprehensive and the paper is well written.

I think this paper can become an important contribution to the public health discussion on the place of UPF in the diet. 

My main concern is related to the separation between the effects of the food additive mixtures per se from the one of the food matrix per se, or even the diet pattern. As mentioned by the authors, the first mixture included additives present in a variety of UPF, including broth dairy desserts, fats and sauces - all foods that have been associated with higher risk of T2D in many cohorts. Likewise, the other group was SSB/ASB driven, which are also important dietary risk factors of T2D (mostly SSB). In an attempt to control for diet quality, authors adjusted for intakes of energy without alcohol (continuous, kcal/d), total saturated fatty acids (continuous, g/d), sodium (continuous, mg/d), dietary fibre (continuous, g/d), alcohol (continuous, g/d), added sugars (continuous, g/d), fruits and vegetables (g/d), dairy products (continuous, ml/day), red and processed meats (continuous, g/d). 

I first question the concomitant presence of nutrients and foods in the model. Adjusting for saturated fat and dairy products and red meat seems inappropriate because these foods are the main sources of saturated fat. I would first suggest using two distinct models, one nutrient-based and the other food-based. 

Even by doing so, the control for the food matrix will likely be imperfect because the food additive mixtures are found in a limited number of different foods. To address this issue, I would invite the authors to test whether the food additive mixtures mediate the relationship between the foods that are the main sources of these food additive and T2D. Such model would provide key information on the potential contribution of the food additive mixture on the causal pathway between the foods that contain these additive and T2D. These models should also include key nutrients know to drive the relationship between the foods and T2D risk. For instance, relative to SSB/ASB consumption, food mixture #5, and T2D: Is food mixture #5 mediating the relationship between SSB/ASB and T2D risk? If yes, how strong is the effect vs with the one of added sugar? If food mixture #5 is not found to be a mediator of the relationship between SSB/ASB and T2D, what are the implications relative to the paper conclusions? The same should be tested with key foods associated with food additive mixture #1.

Also, I would strongly recommend adding analyses stratified by diet quality. The two food additive mixtures associated with T2D are associated with UPF, and UPF consumption correlate with lower diet quality, it appears important to evaluate whether the detrimental relationship between the food additive mixtures and T2D remains valid at any level of diet quality. Such data are key for public health policies.

I acknowledge that my recommendations require significant statistical work, but my concerns echo to key questions in the public health debate on the place of UPF in the diet. Addressing these questions would allow significant advances in the UPF field. 

Merci

Reviewer #3: The paper is quite interesting, and the authors have conducted a remarkable research. However, I have two minor points regarding the study's limitations:

1) The authors stated: "Next, the generalizability of our findings may be influenced by the cohort's demographic characteristics, such as a higher proportion of women and a more health-conscious population. Therefore, caution is warranted when extrapolating our results to broader populations." As the study aims to evaluate an association, the generalizability is not necessarily compromised by demographic characteristics (as in a prevalence st. If the authors believe there are specific biases affecting the associations, they should clearly identify these issues and explain how they might impact the results. 

2) Regarding the final limitation: "Last, this study did not allow us to specifically investigate mechanistic synergies and/or antagonisms between the food additive chemicals characterizing the mixtures per se.". While the authors suggest experimental studies to address this limitation, it's worth noting that certain epidemiological statistical analyses can account for synergistic effects among variables. It would be beneficial if the authors could: a) Explain why this limitation is relevant to the current study. b) Suggest alternative epidemiological approaches that could provide deeper insights into these associations.

---

* Please upload any figures associated with your paper as individual TIF or EPS files with 300dpi resolution at resubmission; please read our figure guidelines for more information on our requirements: http://journals.plos.org/plosmedicine/s/figures. While revising your submission, please upload your figure files to the PACE digital diagnostic tool, https://pacev2.apexcovantage.com/. PACE helps ensure that figures meet PLOS requirements. To use PACE, you must first register as a user. Then, login and navigate to the UPLOAD tab, where you will find detailed instructions on how to use the tool. If you encounter any issues or have any questions when using PACE, please email us at PLOSMedicine@plos.org.

* FINANCIAL DISCLOSURES: The funding statement should include: specific grant numbers, initials of authors who received each award, URLs to sponsors’ websites. Also, please state whether any sponsors or funders (other than the named authors) played any role in study design, data collection and analysis, the decision to publish, or preparation of the manuscript. If they had no role in the research, include this sentence: “The funders had no role in study design, data collection and analysis, decision to publish, or preparation of the manuscript.”

* COMPETING INTEREST: All authors must declare their relevant competing interests per the PLOS policy, which can be seen here: https://journals.plos.org/plosmedicine/s/competing-interests

For authors with ties to industry, please indicate whether any of the interests has a financial stake in the results of the current study.

* DATA AVAILABILITY: Please include the details provided under "Data and Resource Availability" in the Data Availability statement in the online submission.

FIGURES AND TABLES

SUPPLEMENTARY MATERIAL

REFERENCES

* Where website addresses are cited, please include the complete URL and specify the date of access (e.g. [accessed: 12/06/2024]).

STUDY TYPE-SPECIFIC REQUESTS

* Abstract: Please include the study design, population and setting, number of participants, years during which the study took place (enrollment and follow up), length of follow up, and main outcome measures.

* Please ensure that the study is reported according to the STROBE (or appropriate STOBE extension) guideline (available from: https://www.equator-network.org/reporting-guidelines/strobe) and include the completed STROBE (or STROBE extension) checklist as Supporting Information. Please add the following statement, or similar, to the Methods: "This study is reported as per the Strengthening the Reporting of Observational Studies in Epidemiology (STROBE) guideline (S1 Checklist)." When completing the checklist, please use section and paragraph numbers, rather than page numbers. 

* For all observational studies, in the manuscript text, please indicate: (1) the specific hypotheses you intended to test, (2) the analytical methods by which you planned to test them, (3) the analyses you actually performed, and (4) when reported analyses differ from those that were planned, transparent explanations for differences that affect the reliability of the study's results. If a reported analysis was performed based on an interesting but unanticipated pattern in the data, please be clear that the analysis was data driven. 

* Please state in the Methods section whether the study had a prospective protocol or analysis plan. If a prospective analysis plan (from your funding proposal, IRB or other ethics committee submission, study protocol, or other planning document written before analyzing the data) was used in designing the study, please include the relevant document(s) with your revised manuscript as a Supporting Information file to be published alongside your study and cite it in the Methods section. A legend for this file should be included at the end of your manuscript. If no such document exists, please make sure that the Methods section transparently describes when analyses were planned, and when/why any data-driven changes to analyses took place. Changes in the analysis, including those made in response to peer review comments, should be identified as such in the Methods section of the paper, with rationale.

---

## [Decision Letter · Decision Letter 2]

31 Jan 2025

Dear Dr Payen de la Garanderie,

Many thanks for re-submitting your manuscript "Food additive mixtures and risk of type 2 diabetes: results from the NutriNet-Santé cohort" (PMEDICINE-D-24-03030R2) to PLOS Medicine. The paper has been seen again by one subject expert and the statistician; their comments are included below and can also be accessed here: [LINK]

Thank you for your detailed response to the reviewers' comments. As you will see, the reviewers are satisfied with your responses to their comments. However, there are a number of remaining comments from the Academic Editor that require further clarification. After discussing the paper with the editorial team, we ask you to carefully address the comments in a further revision. We plan to send the revised paper to some or all of the original reviewers.

We ask that you submit your revision by Feb 20 2025. However, if this deadline is not feasible, please contact me by email, and we can discuss a suitable alternative.

Don't hesitate to contact me directly with any questions (atosun@plos.org). 

Best regards, 

Alexandra 

Alexandra Tosun, PhD 

Associate Editor

PLOS Medicine

atosun@plos.org

Comments from the academic editor:

The authors addressed reviewers' comments provided previously, but some concerns remain.

Two major comments are supplied hereafter, followed by minor comments.

1.

The authors failed to demonstrate the interactions/synergisms well. The authors tested statistical interactions, showing p-values in eTable 11. Significant interactions do not mean synergism because antagonism between two particular additives could happen. The authors should display the estimates of regression coefficients (+ 95% CI) of two single terms and an interaction term, three in total, for each test for presenting eTable 11. Each food additive may be studentized to mean=0 and SD=1 for interpretability, after the authors preset means and SDs of the food additives.

The tests for interaction miss the directionality. That is one of the issues in the current presentation. Second, the authors found a lot of null findings for interactions. It is unclear if the interaction effects explain the large fraction of the observed associations. Given the exploratory nature of the interaction following the significant testing of each cluster, the authors should tone down their inferences about interactions/synergisms, at least in the abstract.

In the abstract, the authors should not use the words "interaction/synergistic effects", as this study did not primarily evaluate the effects but evaluated the potential effect of a combination or cluster of multiple correlated additives and then explored the interaction.

The following revisions should be made in the abstract:

Background:

The following sentence should be revised: "These findings suggest that potential interaction/synergistic effects of food additives should be considered in safety assessments..."

A revised sentence may be said, "These findings suggest that potential effects of multiple food additives should be considered in safety assessments..."

Results:

The authors should state that several interactions were indicated in their exploratory analyses for one of the clusters, with clear information on the directionality.

Discussion:

The authors stated, "This study revealed positive associations between exposure to two widely consumed food additive mixtures and increased type 2 diabetes risk. Further experimental research is needed to depict underlying mechanisms. These findings suggest that potential interaction/synergistic effects of food additives should be considered in safety assessments, and they support public health recommendations to limit non-essential additives."

Grammer is not right, partly. This sub-section should be revised, "This study revealed positive associations between exposure to two widely consumed food additive mixtures and higher type 2 diabetes incidence. Further experimental research is needed to depict underlying mechanisms, including potential interactions/synergistic effects. These findings suggest that a combination of food additives should be considered in safety assessments, and they support public health recommendations to limit non-essential additives."

2.

This cohort seemed to be based on participants across France. The authors should clarify the geographical diversity and how they addressed the potential confounding due to geosocial factors.

Some cities are by the Mediterranean Sea, while other cities are mountainous, by the Atlantic Ocean or by the Strait of Dover. Cultural diversity was present, as well as racial and geosocial diversity.

Despite that, the authors adjusted only for education levels and professional occupation as socioeconomic factors. The degree of residual confounding owing to the nationwide diversity may be substantial.

The limitation section appears to include the limited opportunity of collecting race/ethnicity/religion. The authors are encouraged to document any such restriction in the method section, so that readers can digest the results with such an important limitation kept in mind.

Minor comments:

Hazard ratios presented in the abstract are not interpretable because of no information on the unit of the exposure. Please clarify.

Line 116-117:

The authors stated, "These associations were not driven by a specific food additive, and several interactions between additives of the same mixture were detected." This interpretation is misleading. There may be single independent effects of several food additives. The effect of an interaction could be minor. Also, the effect of an interaction could be antagonistic, being outweighed by positive independent single effects. As implied above, the authors should give a sentence that implies multiple interpretations.

Line 127-128: It sounds too early to say "should be", given the presentation of eTable 11 with many negative results and without any replication or biologically plausible explanations. The authors may rephrase, "the potential interactions/synergism may be of interest in future mechanistic investigations and considerations in safety assessment".

The authors also have an issue with measurement errors that are common in any observational research of this kind, owing to the dietary assessment undertaken and food-additive databases used. As the authors noted in the discussion section, those limitations must have mattered. Accounting for multiple epidemiological issues, the authors should follow the suggestions given above.

Line 134: "a major hypothesis" should be rephrased to "one of the hypotheses". There is no evidence that the authors' argument is right.

Line 160-163:

After this statement of the primary aim, the authors may want to state their exploratory, secondary aim to examine interactions between food additives in a cluster associated with type 2 diabetes incidence.

Line 170:

If the recruitment was ongoing, there should be a calender time at which the authors formed a cohort for this specific study. For example, it is unlikely that the authors included participants recruited when this manuscript was being written up. The authors should clarify the calendar time when the participants in this specific study were recruited.

Line 188: "are" should be "were".

Line 189: "validated" should be taken out. There is no possibility that any measures of a dietary assessment tool are perfectly valid. What is appropriate is to present how valid main dietary variables would be for the specific hypothesis to test. In this study, the authors failed to provide such information, so "validated" should be taken out, and some degree of validity should not be indicated in this manuscript.

Line 228: "is" should be "was".

Line 341: "risk" should be incidence. The authors should review their text not to misuse risk and incidence. The authors evaluated time-to-event data and analyzed incidence, not risk. It may be acceptable sometimes to use "risk" when inferring a risk. However, when the authors document results derived from their analyses of incidence (N cases / person-time), they should use properly the term, "incidence". It would be shocking if the authors do not understand the difference between risk and incidence.

Line 358-359:

This description should be elaborated more in this text. The authors are interested in interactions/synergisms, but the descriptions of the resulst are overly simple, and no one can judge the strength of evidence or identifiy the numbers of null results and directionality of the potential interactions.

Line 387:

"supporting potential synergistic effects". As mentioned above, the directions of associations were not presented; it is unclear if synergisms were reasonably indicated. The authors should present results sufficiently so that the authors' arguments make sense.

Line 413-415: These hypotheses were not given in the introductory section. At the very least, the brief intention should be documented as suggested above. It is odd to see the authors clarify their hypothesis in the discussion section.

Line 415-419: Confirm those with the directions of the interaction estimates.

Line 449: "highlighting" should be rephrased to "indicating". As repeatedly implicated, the presence of an effect of a specific mixture does not necessarily mean an interaction/synergism.

Line 511: Misclassification of diabetes cases as non-cases, considering undiagnosed diabetes as non-cases, is likely to depend on socioeconomic status and health consciousness. Also, if the size of cohort is big, non-differential misclassification can be overwhelming. The authors should eliminate the argument as if the impact was negligible. (Basically, the authors misunderstood the nature of outcome misclassification. Some outcome misclassifications, such as low positive predictive value, may cause just a power issue, but not necessarily the case).

Line 517-518: Too weak argument. Take it out. Consistency with national averages should not happen when the assessment was valid, as this study recruited more women than men, for example. The authors do not need to supply weak arguments as no study is perfect anyway.

Comments from the reviewers: 

Reviewer #1: The authors have addressed my points.

Michael Dewey

Reviewer #2: Thank you for addressing carefuly the comments raised previously. I have no further comments.

Requests from Editors:

TITLE

Please check the formatting of your title (Is there a line break following the colon?). Please check that the title is according to PLOS Medicine's style. Your title must be nondeclarative and not a question. It should begin with main concept if possible. "Effect of" should be used only if causality can be inferred, i.e., for an RCT. Please place the study design ("A randomized controlled trial," "A retrospective study," "A modelling study," etc.) in the subtitle (i.e., after a colon).

ABSTRACT

1) l.59ff: Please replace ‘y’ with ‘years’ or define ‘y’ at first use. We would prefer the first option.

2) l.59ff: Please define ‘SD’ at first use. Throughout the abstract and main text, please ensure that abbreviations, including statistical abbreviations, are defined the first time they are used.

3) ll.59-60: For clarity, we suggest changing the sentence to: “Participants (n=108,643, mean follow-up = 7.7 years (standard deviation (SD) = 4.6), age = 42.5 years (SD = 14.6), 79.2% women) were adults from the French NutriNet-Santé cohort (2009-2023).”

4) l.68: When reporting 95% CIs please separate upper and lower bounds with commas instead of hyphens as the latter can be confused with reporting of negative values. Please revise throughout the manuscript. Also, please remember to introduce abbreviations, such as ‘HR’.

5) We feel that for full transparency you should also report the results, i.e. HR values, for the three food mixtures that did not show an association with increased diabetes risk.

6) In the last sentence of the Abstract Methods and Findings section, please describe the main limitation(s) of the study's methodology.

7) Please ensure that all numbers presented in the abstract are present and identical to numbers presented in the main manuscript text.

9) Please include the important dependent variables that are adjusted for in the analyses.

AUTHOR SUMMARY

1) Please remove any numerical results from the Author Summary.

2) Please temper claims of primacy of results by stating, "to our knowledge" (or something similar) or remove ‘is the first’.

3) We suggest changing the order of the bullet points under ‘What do these findings mean?’: 

• The study results suggest that food additives found in a wide variety of products and frequently consumed together may potentially represent a modifiable risk factor for type 2 diabetes prevention. They support public health recommendations to limit non-essential additives.

• Potential interaction/synergistic effects of food additives should be considered in future safety assessments.

• Confirmation by other epidemiological and experimental studies will be necessary to support causality of the observed associations.

4) Please note that in the final bullet point of ‘What Do These Findings Mean?’ the main limitations of the study should be included in non-technical language (i.e. the above might require further changes).

INTRODUCTION

1) If there has been a systematic review of the evidence related to your study (or you have conducted one), please refer to and reference that review and indicate whether it supports the need for your study.

2) l.136ff: Please ensure to provide references.

3) ll.153-155: We suggest changing to: “Another important gap to date has been that previous evaluations have not been able to account for potential interaction/synergistic effects when assessing the safety of additives due to a lack of data.”

4) l.155: Please use the abbreviation 'UPF' instead of 'ultra-processed foods' as you introduced the abbreviation early in the Introduction. Please revise throughout.

5) ll.158-160: Please revise for grammar/clarity.

6) l.161: Please temper claims of primacy of results by stating, "to our knowledge" (or something similar).

7) l.163: We suggest writing '...using the large prospective NutriNet-Santé cohort'.

METHODS AND RESULTS

1) ll.223-224: We suggest providing a brief explanation of ‘under-energy reporters’ (e.g. in parenthesis).

2) l.251: Please spell out ‘T1D’ as you have not introduced the abbreviation.

3) l.295: The terms gender and sex are not interchangeable (as discussed in https://www.who.int/health-topics/gender#tab=tab_1 ); please use the appropriate term (also in the Abstract).

4) l.295, we suggest changing this to "At baseline, the median age of the cohort was ..." to avoid that "their" refers only to females.

5) ll.298-299: “Participants completed a median of 5 dietary records (25th – 75th percentiles: 3-9).” – does this refer to the entire NutriNet-Santé cohort or to the participants specifically included in this study? Please revise the characteristics paragraph for clarity.

6) ll.324-317: We suggest splitting the sentence into two for clarity.

7) l.332: Please remove the word ‘famous’.

8) l.345: If you agree, we feel it is worth repeating here (briefly) the types of sensitivity analyses that have been carried out.

9) l.353ff: Your study is observational and therefore causality cannot be inferred. Please remove language that implies causality, such as effect. Please refer to associations instead. Please revise throughout (including the discussion).

10) l.354: Please note that you have not yet introduced the abbreviation 'T2D' and we would suggest that you continue to write 'type 2 diabetes'. Please revise throughout.

11) Table 1: Please ensure that all abbreviations below the table, such as BMI and SD, are defined. Please also refer to comment 3) and revise accordingly (sex versus gender).

12) Table 2: Please indicate the meaning of the values following the food additive in the table heading.

13) Figure 1: Please include a definition of the five mixtures in the figure description. Please be sure to define all abbreviations (SD, BMI, CI, etc.). Please add a title for the x-axis. Please replace the hyphens with commas.

14) Figure 2: The image quality is very poor and, for example, the y-axis title is barely readable. Please revise the axes titles. Please also add a definition of mixture 2 and 5.

15) Figures 1 and 2 appear rather small at the moment. Please revise, also with regard to the comment about image quality.

DISCUSSION

General guidance: Please present and organize the Discussion as follows: a short, clear summary of the article's findings; what the study adds to existing research and where and why the results may differ from previous research; strengths and limitations of the study; implications and next steps for research, clinical practice, and/or public policy; one-paragraph conclusion.

1) ll.431-434: Please ensure to indicate that these research findings result from experiments in mice. Please revise throughout.

2) l.443: “type 2 diabetes patients” - PLOS Medicine prefers the use of patient-centered language, e.g. patients with type 2 diabetes. Please revise throughout.

3) l.522: : Please temper claims of primacy of results by stating, "to our knowledge" (or something similar).

4) Please remove any subheadings.

REFERENCES

1) Where website addresses are cited, please use the word ‘accessed’ when specifying the date of access (e.g. [accessed: 12/06/2024]).

2) Please make sure to check and update the references where necessary (e.g. [9]).

SUPPLEMENTARY MATERIAL

1) Please ensure that all supplementary files are referenced in the main text.

2) In the published article, supporting information files are accessed only through a hyperlink attached to the captions. For this reason, you must list captions at the end of your manuscript file. You may include a caption within the supporting information file itself, as long as that caption is also provided in the manuscript file. Do not submit a separate caption file.

When SI files are contained with a single file:

Please label the file as ‘S1 Supporting Information’.

Please apply alphabetical labelling to each table and figure contained within the S1 file. For example, ‘Fig A’ to ‘Fig Z’ and ‘Table A’ to ‘Table Z’.

Plain text does not need to be labelled and can just be given a title as necessary. For example, ‘Statistical Analysis Plan’.

Please cite tables/figures as ‘Fig A in S1 Supporting Information’ and/or ‘Table A in S1 Supporting Information’, for example.

Please cite plain text as, ‘Statistical Analysis Plan in S1 Supporting Information’, for example.

When SI files are uploaded as separate files:

Please label tables as ‘S1 Table’ (so on) and figures as ‘S1 Fig’ (and so on).

Any additional documents (protocols/analysis plans etc.) can be labelled as ‘S1 Protocol’, for example. Please cite items as exactly as labelled.

General Editorial Requests

---

## [Editor Report · Decision Letter 3]

28 Feb 2025

Dear Dr Payen de la Garanderie, 

On behalf of my colleagues and the Academic Editor, Fumiaki Imamura, I am pleased to inform you that we have agreed to publish your manuscript "Food additive mixtures and type 2 diabetes incidence: Results from the NutriNet-Santé prospective cohort" (PMEDICINE-D-24-03030R3) in PLOS Medicine.

I appreciate your thorough responses to the reviewers' and editors' comments throughout the editorial process. We look forward to publishing your manuscript, and editorially there are only a few remaining minor stylistic points that should be addressed prior to publication. We will carefully check whether the changes have been made. If you have any questions or concerns regarding these final requests, please feel free to contact me at atosun@plos.org.

Please see below the minor points that we request you respond to:

1) Table 1: Please change the baseline characteristic 'Women' to 'Sex, Female'.

2) Table 2: Please add footnote 'a' below the table before 'The loading values...'.

3) Figure 2: The image is still quite small. Please define the meaning of the dashed line in the image description (define 'CL').

Before your manuscript can be formally accepted you will need to complete some formatting changes, which you will receive in a follow up email (including the editorial points above). Please be aware that it may take several days for you to receive this email; during this time no action is required by you. Once you have received these formatting requests, please note that your manuscript will not be scheduled for publication until you have made the required changes.

PRESS

Sincerely, 

Alexandra Tosun, PhD 

Associate Editor 

PLOS Medicine